# The Impact of the Gain-Loss Frame on College Students’ Willingness to Participate in the Individual Low-Carbon Behavior Rewarding System (ILBRS): The Mediating Role of Environmental Risk Perception

**DOI:** 10.3390/ijerph191711008

**Published:** 2022-09-02

**Authors:** Ani Qi, Zeyu Ji, Yuanchao Gong, Bo Yang, Yan Sun

**Affiliations:** 1Key Laboratory of Behavioral Science, Institute of Psychology, Chinese Academy of Sciences, Beijing 100101, China; 2Department of Psychology, University of Chinese Academy of Sciences, Beijing 100049, China; 3Economy School, Zhengzhou University of Aeronautics, Zhengzhou 450015, China

**Keywords:** framing effect, nudge, environmental risk perception, Individual Low-Carbon Behavior Rewarding System (ILBRS)

## Abstract

Since Chinese households account for more than half of the country’s total carbon emissions, efforts focused on consumption will be key to reaching carbon reduction targets. The Individual Low-carbon Behavior Rewarding System (ILBRS) is an emerging mechanism in China that encourages the public to develop a low-carbon lifestyle and it is critical to look for various approaches to enhance the public’s willingness to participate in it. The framing effect has been widely used to study pro-environmental behavior as a low-cost nudge. We used an online questionnaire (N = 320) to investigate how framing information (loss and gain framing) influenced people’s willingness to participate in the ILBRS through the mediation of environmental risk perception. The results indicated that the public’s willingness to participate in the ILBRS under the loss frame was significantly higher than the gain frame. Furthermore, environmental risk perception played a mediating role in the proceedings. Based on our findings, the designers and promoters of ILBRS systems could employ loss-frame information to promote the public’s willingness to participate in the ILBRS and drive more people to live a low-carbon life in the process of mechanism construction, information communication, and operational promotion.

## 1. Introduction

As the world’s largest emitter of greenhouse gases, China is facing a considerable challenge [1]. To reduce carbon emissions, the Chinese government has developed various policies, such as carbon trading and green energy policies. Many studies agree that these policies have caused a significant impact on carbon emission reduction on the production side [2,3,4]. However, from the perspective of the entire life cycle of products, consumers are the primary driver of product transformation [5]. According to statistics, the greenhouse gas emissions caused by household consumption in China already account for 52% of the total national emissions [6]. Based on the experience of developed countries, with the living standard improving, the proportion of domestic energy use will rise and eventually exceed that of industrial energy [7]. Therefore, it is significant to reduce emissions on the consumption side, and lead the public to practice low-carbon living. China’s Individual Low-Carbon Behavior Rewarding System (ILBRS) is an emerging localized system that integrates the theories of economics, management, behavioral science, environmental psychology, and environmental education. It encourages small and micro-enterprises, community households, and individuals to participate in low-carbon behaviors through voluntary participation and behavior records, and then uses quantitative data accumulation to establish an incentive mechanism. The system aims to guide society to participate in green and low-carbon development [8]. The ILBRS operating mechanism calculates carbon emissions and emission reductions through scientific methods by registering them in personal carbon credit accounts and converting them into carbon credits using coefficients, which can be exchanged for goods and coupons at relevant equity platforms. There are many related implementation examples, such as the government-sponsored “Green Travel” in Beijing, “Carbon Planet” in Wuhan, “TanHuiTianFu” in Chengdu, “Low Carbon Planet” in Shenzhen, and the enterprise-sponsored “Ant Forest”.

Many ILBRS projects have achieved tremendous things in recent years. According to the related reports, Fuzhou City, Jiangxi Province, established an ILBRS platform in 2017. The number of registered users reached 400,000 in four years, distributing more than 73.71 million carbon coins and reducing carbon emissions by about 160,000 tons [9]. In September 2020, the green travel ILBRS-MaaS (Mobility as a Service) project, launched by Beijing Municipality in conjunction with Gaode Map and other APPs, reported 420,000 monthly active users in only six months after its launch. More than one million users registered, with a cumulative carbon emission reduction of nearly 100,000 tons, equivalent to 100,000 fuel-fired cars being taken off the road [10]. In July 2016, the Ant Forest was launched on the Alipay platform. By the end of 2020, it had exceeded 500 million users since its inception, with a cumulative emission reduction of 7.92 million tons [11]. The ILBRS has covered various fields, including clothing, food, housing, and transportation through the projects involving green travel, suspension of vehicle stops, garbage classification, water and electricity savings, green consumption, and recycling of used items [6]. These results imply that the ILBRS has excellent value and potential for emission reduction.

Nevertheless, it still fails to achieve the purpose of widely guiding the public to live a low-carbon life with low public participation. Due to the high operating costs and insufficient publicity, some ILBRS platforms have lost users and ceased operation. Existing studies generally discuss the system design and practical significance of the ILBRS, but few discuss how to increase public willingness to participate in the ILBRS from a psychological perspective [12,13,14]. Therefore, it is necessary to study how to increase the public’s willingness to participate in the ILBRS.

Nudge is recognized as an effective tool that can coordinate behaviors and intentions to promote behavioral change. It has been used by several governments and organizations around the world to improve public policies on the environment, health, and education [15,16,17,18,19]. It optimizes public policy implementation by designing decision elements such as makers, decision options, decision processes, and decision environments to influence decision-makers’ preferences and change decision outcomes. Its implementation is simple, low cost, and non-coercive [20,21].

Positive and negative framing measures (framing effects) are an essential nudge method. The framing effect was first introduced by Tversky (1981) and led to different decision judgments by makers through different descriptions of an identical problem [22]. Existing studies have shown that the framing effect arises due to intuitive experiences and emotional preferences (e.g., loss aversion). Different gain and loss framings influence people’s decisions in the environmental field [23]. To date, framing effects have been used to study energy conservation and greenhouse gas emissions [24,25,26,27]. The ILBRS mechanism is closely related to environmental protection and energy saving. Moreover, its content lends itself to different framing designs, such as the description of the usefulness of energy accumulation in Ant Forest. However, there is no research with which to explore and optimize the design of the ILBRS from the perspective of framing effects. Therefore, we explore whether the wording of different framings affects the public’s willingness to participate in the ILBRS.

Different framing wordings elicit different emotional responses, and the emotion is an important factor influencing risk perception [28]. Risk perception refers to an individual’s subjective feelings and perceptions of various objective risks in the external environment [29]. It has been shown that frame manipulation can change environmental risk perceptions and that risk perceptions affect people’s behavioral intentions [30]. Therefore, this paper adds environmental risk perception as a mediating variable to investigate the role of environmental risk perception in the influence of gain and loss framing on individual willingness to participate in the ILBRS and provide a reference for the optimization of the ILBRS.

### 1.1. Operating Status of ILBRS

Individuals are the drivers of green transformation in society and a vital participating force in the carbon market. Many countries have explored individual energy-saving and emission reduction mechanisms, such as South Korea’s Green Card, Japan’s environmental point system, and China’s ILBRS [31]. However, the theoretical research of the ILBRS is still in the initial stage. The ILBRS aims to establish a set of the long-term credit system and adopt the approach of quantitative data accumulation to encourage and benefit public welfare low-carbon behaviors, and then promote and innovate the development of the carbon market. The core of the ILBRS is to adopt an incentive approach to guide energy saving and emission reduction by quantifying the public’s low-carbon behavior. Some ILBRS platforms have gained widespread public attention and participation, such as the Puhui Certified Emission Reductions in Guangdong Province and the Ant Forest platform. However, due to the limited operating cost, most of the ILBRS platforms use rewards of low-value goods to motivate customers to participate, such as a bottle of mineral water and a small coupon. Some of the ILBRS programs have even ceased operation due to their inability to provide sustainable financial incentives, such as TanBaoBao, launched in 2016 in Wuhan, Carbon Account, launched in 2015 in Shenzhen, and ZaoDianXingQiu, launched in 2019 in Chengdu. These programs often required significant financial support during promotion and were eventually discontinued due to low attractiveness to the public, unsustainability, and inconvenience.

### 1.2. Framing

Low material incentives cannot fully motivate public participation, while long-term financial incentives increase the burden on the platform or the government. We, therefore, introduce a non-priced nudge approach—the framing effect—to investigate the facilitating effect of framing information on increasing public participation in the ILBRS. Tversky (1981) discovered and proposed the framing effect in a study on Asian disease problems. In the experiment, participants preferred to choose the risky option in the negative frame and the deterministic option in the positive frame. Tversky argued that differences in the form of expression caused different choices. Framing effects are typically classified as risky choice framing effects, feature framing effects, and goal framing effects [29,30,32]. In environmental decision-making studies, 93% used goal framing [25]. Goal framing refers to using different statement frames for the goal of an action or behavior, mainly describing the relationship between the behavior and the goal in terms of performing or not performing a behavior. For example, Spence and Pidgeon (2010) described the gain frame as follows: “By mitigating climate change, we can prevent further increases in winter floods in maritime regions and flash floods throughout Europe” and the loss framing as follows: “Without mitigating climate change, we will see further increases in winter floods in maritime regions and flash floods throughout Europe” in their study of the framing effect on public perceptions of the severity of climate change [33].

Regarding the impact of the gain/loss framing, current research has yielded mixed results. Some studies have shown that loss frames are more effective in influencing behavior. For example, loss frames have a more significant effect on promoting recycling behavior than gain frames [34]. A comparison of 61 framing studies on the environment found that loss framing was more effective in influencing behaviors and intentions [25]. Studies have shown that loss framing can promote action by eliciting anxiety and fear [35]. Furthermore, a study on the issue of vaccination of newborns noted that loss framing was more likely to promote people’s behavior and intentions during events with some risk [36]. It may be related to persons’ loss aversion, in which people are more averse to the same loss relative to the gain. In the relevant context of promoting pro-environmental behavior, the loss frame that emphasizes the consequences of inaction is more convincing than the gain frame that emphasizes action [37]. Researchers attribute these results to the perception that negative messages are more vivid, more intimidating, and more important than positive messages. In a world dominated by positive information, negative information may be more prominent in people’s perceptions. Compared with the gain frame of the ILBRS, the content of the loss frame of the ILBRS may be more likely to increase people’s behavioral intention to participate in the ILBRS via causing negative emotions, such as anxiety and fear about environmental degradation, and bringing more loss averse people. Therefore, we propose here the first hypothesis of this study.

**H1:** 
*Individuals’ willingness to participate in the ILBRS in loss framing is significantly higher than in the gain framing.*


### 1.3. Environmental Risk Perception

Risk perception refers to the individuals’ subjective feelings and perceptions about various objective risks in the external environment. It was initially introduced by Bauer (1960) in consumer behavior [38]. Risk perception is an integral part of risk research and has received much attention from scholars [39], and Slovic (1987) noted that most of the public judge risk through intuition and personal experience [40]. Environmental risk perception is the subjective feeling and perception of risk in the natural environment. Numerous studies have shown that framing can significantly influence risk perception, and scholars have generated different perspectives [41,42,43]. Since 1960, efforts have been made to highlight the dangerous consequences of inaction in a fearful and catastrophic portrayal to elicit public perceptions of environmental risk. For example, Stephen Hawking wrote in The Times that terrorist attacks would only kill hundreds of people while climate warming could kill millions [43]. Related studies have also shown that loss frames are more fear inducing and thus enhance people’s perceptions of environmental risk [33]. In our view, as suggested in the previous literature, the negative way of communicating the risks present in the environment can induce an increase in public awareness of environmental risks and concerns to a greater extent, which may be related to people’s loss aversion. Therefore, we argue that loss framing can elicit higher risk perceptions. In addition, environment-related studies have shown that environmental risk perception affects people’s willingness to engage in pro-environmental behaviors [44]. Accordingly, we infer that loss framing increases environmental risk perceptions more than gain framing. Higher environmental risk perceptions can promote people’s willingness to engage in the ILBRS, and we propose the following hypothesis.

**H2:** 
*Loss framing significantly increases environmental risk perceptions compared to gain framing.*


**H3:** 
*Environmental risk perception plays a mediating role in the framing’s effects on people’s willingness to participate in ILBRS, and the loss frame causes more environmental risk perception to increase the individuals’ intentions to engage in the ILBRS.*


## 2. Materials and Methods

### 2.1. Participants

According to previous studies, educational attainment, personal economic background, gender, and age influence environmental risk perceptions [45]. Therefore, by selecting college students, we were able to control irrelevant variables to some extent, such as education level, individual income, and age. The questionnaires were distributed with different framing information to the college student group on the WeChat platform. WeChat is a free social application in China available for smartphones that lets you message and call friends and family, and share photos, videos, documents, and voice messages. The extreme value samples with monthly living expenses of less than 500 yuan (N = 2; 0.6% of the total sample), the education level of doctoral students (N = 3; 0.9% of the total sample), and invalid questionnaires were excluded. These extreme value samples represented a very small percentage of the total sample, and we checked that removing these samples did not affect the results of our final data analysis. Finally, 320 valid questionnaires were included in the statistics. The average age of the subject group was 24.7 years old. Specific demographic information is shown in Table 1.

### 2.2. Experimental Procedure

This study used a between-participants design to provide participants with different textual framings (loss framing description or gain framing description). Then we measured the participants’ willingness to participate in the ILBRS and environmental risk perceptions to explore the influence of loss and gain framing on the willingness to participate in the ILBRS and the mediating role of environmental risk perceptions. The process details were as follows: Participants received one of the two questionnaires in WeChat. They were required to read the introduction of the ILBRS under the gain framing (or loss framing) first, then the level of environmental risk perception was measured, and finally, they answered the questions on demographic-related variables. The reading time of the material was limited to at least 15 s to ensure the participants adequately read the preceding frame material.

We used SPSS statistical software and the PROCESS program to organize and analyze the data. First, descriptive statistics were performed. An independent sample t-test was used to analyze the effect of the gain and loss framing on willingness to participate and environmental risk perceptions. Finally, the PROCESS program was used to mediate the data using the Bootstrap method.

### 2.3. Experimental Method

The material for introducing the ILBRS learns from the basic introduction of the ILBRS platforms such as Ant Forest and Carbon Planet. The descriptions of the gain and loss framings refer to the study by Spence (2010) [33]. We describe the framing information as follows.


**Gain frame:**
Collecting 2280 g of carbon energy means we can protect 1 square meter of the primeval forest which is the home of animals.Collecting 10,800 g of carbon energy means we can protect 10 square meters of Sanjiangyuan Park which is the home of the Chinese snow leopardCollecting 20,000 g of carbon energy means that one sand willow can be planted in the desert to stop the spread of the Mongolian desert by 1 square meterCollecting 50,000 g of carbon energy means that one camphor pine can be planted to help absorb 50 g/day of CO_2_ and slow down climate change.Collecting 80,000 g of carbon energy means we can plant an elm tree, stopping 1 square meter of arable land in the Yellow River basin from being washed away.



**Loss frame:**
Without collecting 2280 g of carbon energy, 1 square meter of virgin forest is destroyed without protection and animals lose their homes.Without collecting 10,800 g of carbon energy, 10 square meters of the Sanjiangyuan Park, home of the Chinese snow leopard, are destroyed due to lack of protection.Without collecting 20,000 g of carbon energy, 1 square meter of the Mongolian desert spreads forward due to the lack of a sand willow.Without collecting 50,000 g of carbon energy, the carbon dioxide we produce today worsens the climate as a result of the lack of a single camphor pine to absorb it.Without collecting 80,000 g of carbon energy, 1 square meter of arable land in the Yellow River basin is washed away by heavy floods due to the lack of a single elm tree to hold it in place.Questions on demographic variables: including gender, age, education, profession, city of residence.


Willingness to participate in the ILBRS: according to the acceptance model, the ILBRS willingness to participate scale refers to Saari’s (2021) behavioral willingness scale [30], which uses a 1–100 scale (1 = strongly disagree, 100 = strongly agree), for example, “I would use the Low Carbon Planet applet”. The five questions of the questionnaire had a high internal consistency (Cronbach’s α = 0.926).

Environmental risk perceptions: the scale of environmental risk perceptions was chosen from the study of Leiserowita (2006) [46]. A question was used to measure the participants’ perception of environmental risk. After reading the material, the participants feel that environmental degradation has severely impacted where they lived (1 = strongly disagree, 100 = strongly agree).

## 3. Results

### 3.1. Descriptive Statistics and Tests of Variance

In this study, questionnaires were distributed through the Internet, and finally 320 valid questionnaires were collected, including 156 in the gain frame. The results of comparing people’s willingness to participate in the ILBRS and environmental risk perceptions under loss and gain framing by independent sample t-test are shown in Table 2. The participants’ willingness to participate in the ILBRS was significantly higher in the loss frame than in the gain frame (*t* = −3.103, *p* = 0.002). The person’s perception of environmental risk was significantly higher in the loss frame than in the gain frame (*t* = −2.259, *p* = 0.025). These results validate hypotheses 1 and 2 of this study.

### 3.2. Hierarchical Regression Analysis

In order to further verify whether the frames significantly affect people’s willingness to participate in the ILBRS, and excluding the effect of other demographic variables on the main effect, a series of multiple regression analyses was conducted. In step 1, we put the demographic variables into the model using the enter method, including gender, age, education level, and monthly living expenses. Based on step 1, we put the independent variable, the frame, into the model in step 2. We obtained the results shown in Table 3. In the model of step 2, gender, age, education, and monthly living expenses did not significantly affect willingness to participate in the ILBRS (*p*s > 0.05). Controlling these variables, we found that the framing still significantly (*p* = 0.021) influenced the dependent variable. The results of the hierarchical regression analysis show that, after controlling for the effect of demographic variables, the independent variable still significantly affected the dependent variable and the main effect was still significant.

### 3.3. Intermediary Model Analysis

A bias-corrected non-parametric percentile bootstrap method was used in SPSS in the process macro-program to further investigate the relationship between the framing, environmental risk perception, and willingness to participate. Moreover, 5000 self-sampling tests were conducted to examine the mediation effect of the environmental risk perception in the framing effect on the willingness to participate in the ILBRS based on the mediation effect test method proposed by Zhonglin Wen et al. (2004) according to the research hypothesis [47]. The results of the mediation effect analysis showed that loss framing could significantly increase people’s environmental risk perceptions. When loss framing and environmental risk perceptions were included in the regression equation, both could increase the willingness to participate in the ILBRS. The bootstrap 95% confidence interval of the mediation effect does not contain 0 ([−7.561, −0.591]), indicating a significant mediation effect. The path diagram of the mediating effect is shown in Figure 1. Thus, the environmental risk perception mediates the framing of information and the willingness to participate in the ILBRS. The proportion of the mediating effect to the total effect is 48.56% (see Table 4 for details).

Combining the results of the t-test and regression analysis, we can learn that the loss frame significantly increased the environment risk perception and thus increased the individuals’ intention to participate in the ILBRS, where the environment risk perception played a mediating role. These results validate hypotheses 3 of this study.

## 4. Discussion

The ILBRS, a vital tool to reduce carbon emissions from the consumer end, can raise the public’s awareness of reducing carbon and environmental protection and guide them in developing low-carbon and green living habits. Many studies explore the mechanism of the ILBRS, but few studies pay attention to the willingness to promote the use of the ILBRS. This paper investigates the effect of framing on the willingness to participate in the ILBRS through an experimental questionnaire. Conducting an independent sample t-test on the resultant data, we concluded that the willingness to participate under loss framing is significantly higher than that under gain framing, which is consistent with hypothesis H1. This indicates that different linguistic descriptions of the framing show different degrees of impact on the willingness to participate in the ILBRS. People are more sensitive to loss framing. This point validates the prospect theory that people are more sensitive to losses than gains when making decisions. This phenomenon may be related to people’s loss aversion, where a certain amount of loss brings more utility reduction than the same amount of gain brings utility increase.

Furthermore, it has been previously shown that perceived risk promotes people to take action when they believe they are capable of acting and can successfully avoid or mitigate adverse outcomes [48]. The ILBRS serves as a dual feedback mechanism to let the individuals know they have the ability to protect the environment, and how much they do, and then promote individuals’ low-carbon behaviors. On the one hand, individuals can be informed of the carbon emissions resulting from low-carbon behavior. On the other hand, participants can know how many negative environmental outcomes can be avoided and saved by exchanging corresponding environmental public welfare projects [49]. At the same time, our experiments show that loss framing leads to high-risk perception. Thus, the ILBRS is more likely to increase willingness to take action under a loss framing. Meanwhile, in the study of framing effects in the environmental field, researchers have also found that loss framing is more effective in an active avoidance framing [25]. Our results are consistent with these findings, suggesting that people are more willing to participate in the ILBRS under loss framing rather than gain framing.

Moreover, according to previous studies, the framing effect is influenced by different factors, such as emotions, information sources, and risk perceptions [25]. Through environmental risk perceptions, the framing effect can affect environmental willingness [30]. Therefore, we further investigated the role of environmental risk perceptions in influencing willingness to participate in the ILBRS under different framing conditions. Firstly, we measured participants’ environmental risk perceptions under different frames. We found that participants perceived higher environmental risks under the loss frame, indicating that the framing effect can significantly affect environmental risk perceptions. Then, we conducted a mediation analysis with environmental risk perception as a mediating variable. The results showed that environmental risk perception partially mediated the framing effect on willingness to participate in the ILBRS and that higher environmental risk perception will significantly increase the willingness to participate in the ILBRS. Its mediating effect accounted for nearly half of the total effect and was a significant mediator.

### 4.1. Theoretical Contributions and Policy Recommendations

Firstly, this paper builds on a psychological perspective to explore how to promote public willingness to participate in the ILBRS. Most studies on the ILBRS nowadays focus on institutional design and other content but rarely explore how to promote public participation from a psychological perspective. This study explores the factors influencing the promotion of public participation in the ILBRS by introducing the boosting framing and finds that loss framing is more effective than gain framing in guiding the public’s willingness to participate in the ILBRS. The results support the prospect theory that the gain and loss frame determine how people perceive information. Because of loss aversion, public willingness to participate in the ILBRS is significantly higher under the loss frame than under the gain frame. Second, our study confirms the critical role of environmental risk perception variables in the ILBRS related research, providing a reference for future research in this area. This study explores the mediating role of environmental risk perceptions in the framing effect on the willingness to participate in the ILBRS. It shows that environmental risk perceptions partially mediate the effect of framing on the willingness to participate in the ILBRS. The indirect effect of 48.56% shows that environmental risk perceptions are an essential mediator of the framing effect on the willingness to participate in the ILBRS. Therefore, subsequent studies can continue to explore the application of the framing effect in environmental policy from the view of risk perception.

From the policy perspective, this study provides new ideas for optimizing and promoting the ILBRS. First, the current ILBRS is mainly promoted with positive message frames, while our study shows that using loss frames in the ILBRS is more effective in increasing public willingness to participate. According to the research in this paper, the ILBRS can be promoted with more negative information in public settings, and the program design of the ILBRS can also emphasize more negative framing in the frame building and feedback mechanism, etc. Second, the ILBRS currently uses economic incentives to attract public participation. This study provides non-economic incentives, nudge, and demonstrates that nudge positively impacts the willingness to participate in the ILBRS. Some researchers suggest that the best results can be achieved by using multiple boosting methods in one setting [50,51]. Thus, we suggest multiple forms of boosting to promote the public’s willingness to use the ILBRS, such as simplifying the ILBRS participation and operation process, and adding fun and multifaceted demonstrations. Finally, we also found that environmental risk perception can positively predict people’s willingness to use the ILBRS, and some researchers found that environmental risk perception can improve people’s environmental behavior [52]. Therefore, improving people’s environmental risk perception is also an effective way to promote people’s willingness to participate in the ILBRS, such as through using images of climate catastrophe in the ILBRS promotion process.

### 4.2. Limitations and Future Research Directions

Although this study has experimentally demonstrated the framing’s effectiveness on the public’s willingness to participate in the ILBRS, there are still some limitations. On the one hand, this study used an online questionnaire to explore the public’s willingness to participate in the ILBRS. It is a complex process from behavioral willingness to taking action. The one who has a high willingness may not necessarily take actual action. On the other hand, the sample in this study was made up of a group of college students, which cannot represent all the objects of the ILBRS. A more diversified sample should be selected in the future to verify its general effect.

At present, more theoretical research and practical innovation are needed to explore the psychological mechanism behind public support for the ILBRS and to explore the innovation of the ILBRS incentive mechanism and publicity content, which will provide more practical ideas for the design of the ILBRS in the future. Therefore, we have several suggestions for future research directions. First, researchers should explore long-term nudge methods because energy saving and carbon emission reduction are ongoing tasks that require constant public participation. Second, researchers should explore more effective nudge tools to help attract more people to participate in the ILBRS. Third, researchers should investigate the influencing factors of public participation in the ILBRS through field experiments. In addition, future researchers should expand the experimental sample to include all social classes and age groups, so that more comprehensive and integrated findings can be obtained. Fourth, researchers can conduct a more comprehensive study on the mechanisms underlying the influence of the framing on individuals’ willingness to participate in the ILBRS. In this study, environmental risk perception failed to play a complete mediating role, suggesting the potential of other influence factors.

## 5. Conclusions

This study investigates the effect of different framings on the willingness of the public to participate in the ILBRS and the role of environmental risk perceptions in it. The results show that willingness to participate in the ILBRS is significantly higher in loss framing than in the gain framing. The perception of environmental risk plays a partially mediating role. Loss framing can influence people’s willingness to participate in the ILBRS by affecting their perception of environmental risk.

Overall, this paper enriches the application of framing effects in the environmental protection field, expands the research to the mechanism of framing effects, and provides a theoretical basis and new ideas for the ILBRS in terms of content construction and policy promotion.

## Figures and Tables

**Figure 1 ijerph-19-11008-f001:**
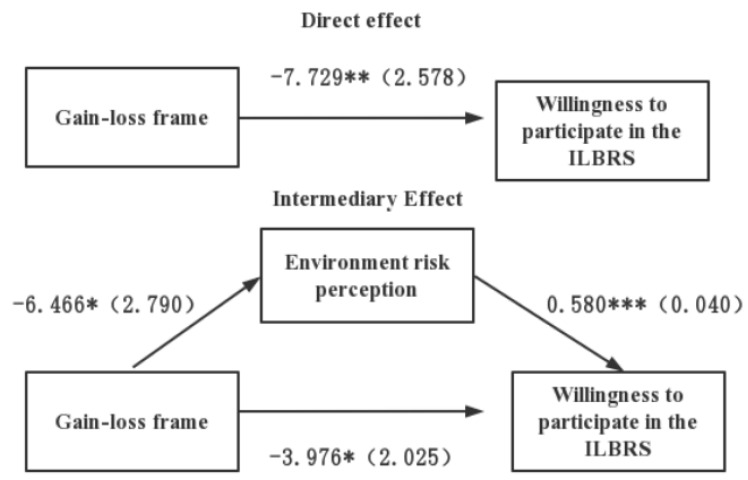
Path diagram of the analysis of the perception of environmental risk on the gain and loss frame on the mediation of the ILBRS participation intention. Note: gain-loss frame: 0 = loss frame, 1 = gain frame; *** *p* < 0.001, ** *p* < 0.01, * *p* < 0.05.

**Table 1 ijerph-19-11008-t001:** Demographic information.

	Variable Level	Sample Size (N = 320)	Percentage (%)
Sex	Male	162	50.6
	Female	158	49.4
Grouping	Group under loss framing	164	51.3
	Group under gain framing	156	49.7
Age	17–25 Years old	228	71.2
	Above 25 Years old	92	28.8
Monthly living expenses	500 RMB–1000 RMB	39	12.2
	1000 RMB–1500 RMB	116	36.3
	1500 RMB–2000 RMB	87	27.2
	2000 RMB–3000 RMB	54	16.9
	Above 3000 RMB	24	7.5
Education	Bachelor Degree	255	79.7
	Specialty	33	10.3
	Master’s Degree	32	10.0

**Table 2 ijerph-19-11008-t002:** Difference test of the willingness to participate in the ILBRS and the perception of environmental risks under the gain and loss frame.

Variables	M ± SD		*t*	*p*
	Loss Frame (N = 164)	Gain Frame (N = 156)		
Willingness to participate in the ILBRS	79.23 ± 19.03	71.50 ± 26.62	2.999	0.003
Environment risk perception	76.79 ± 21.99	70.33 ± 27.72	2.317	0.021

**Table 3 ijerph-19-11008-t003:** Hierarchical regression analysis of the willingness to participate in the ILBRS.

Variables	Step 1	Step 2
Intercept	91.787 *** (11.285)	94.607 *** (11.273)
Gender	−0.105 (2.624)	−0.083 (2.643)
Age	0.017 (0.423)	0.010 (0.421)
Education	−0.151 ** (2.918)	−0.131 (2.932)
Monthly living expenses	−0.025 (1.168)	−0.044 (1.174)
Frames		−0.133 * (2.672)
*F*	3.291 *	3.743 **
*R* ^2^	0.040	0.056
Δ*R*^2^	0.040	0.016

Note: Willingness to use ILBRS application is the dependent variable. Values are standardized regression coefficients, with standard errors in parentheses; *** *p* < 0.001, ** *p* < 0.01, * *p* < 0.05.

**Table 4 ijerph-19-11008-t004:** Analysis of mediating effects of environmental risk perception.

Effect	Effect Value	Boot Standard	Low Limited of 95%	Upper Limited of 95%	Ratio of Total Effect
Total effect	−7.729	2.578	−12.800	−2.658	-
Direct effect	−3.976	2.025	−7.960	0.008	-
Indirect effect	−3.753	1.772	−7.561	−0.591	48.56%

## Data Availability

All data were uploaded on the Figshare. Other researchers can download the dataset at https://figshare.com/s/04ec625ef859f4130580 (accessed on 6 July 2022).

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
