# Peer review of "The Impact of the Gain-Loss Frame on College Students’ Willingness to Participate in the Individual Low-Carbon Behavior Rewarding System (ILBRS): The Mediating Role of Environmental Risk Perception"

_ijerph, 2022, doi:10.3390/ijerph191711008_

Round 1

Reviewer 1 Report

The article concerns a very important contemporary issue of pro-ecological behavior and special attention was paid to the behavior leading to the reduction of carbon dioxide emissions. The individual low-emission behavior reward system (ILBRS) operating in China was presented. This mechanism should encourage society to develop a low-carbon lifestyle and it is important to look for different tools to increase society's readiness for carbon-reducing behavior.

The research goal was correctly formulated. Three research hypotheses have been identified, but the content of the first and third are too synonymous. I propose to make the third hypothesis more specific and not to repeat the content of the first hypothesis. I also did not find a precise indication in the text which results verify the third hypothesis.

It is not necessary to separate chapters 2.3.1 and 2.3.2. I propose to keep only 2.3 "Experimental method".

I cannot find any justification for the conclusion regarding the promotion of the information. However, I agree that the article provides theoretical foundations and new ideas for the construction of a research tool.

I share the opinion that it would be good to extend the study to other age groups and conduct it in the long term. Promotion of this type of solutions is also extremely important, and the article itself is only information that reaches a small part of the society.

Reviewer 2 Report

Introduction:

- I feel that readers have enough information about ILBRS on page 3. The details described in 1.1 Operating Status of ILBRS could be summarized and moved there; Table 1 does not seem necessary.

- p.5, l.149-151: Given framing is at the heart of the paper, it would be relevant to explain why gain frames are generally more effective than loss frames (especially since the authors make the opposite assumption). Related to that, I do not understand the explanation provided by the authors on page 5 (l.179-184). It appears that previous studies showed that a gain frame is more effective than a loss frame to trigger environmental risk perception; the authors attempt to explain why the expect the opposite, but their argument is unclear (to me)

Materials & Methods

- p.6, l.201: Could it be possible to briefly describe the WeChat platform?

- p.6. l.202: Why excluding respondents with very low living expenses? Are the results similar if these respondents are still included (I guess so, but it would be informative to describe it)?

- T-tests do not allow including control variables. I suggest using regression analyses (especially since PROCESS is based on these analyses) to test for the main effect of the frame. By doing so, gender could be controlled for (men and women are known to react differently to gain vs. loss frame; e.g., https://www.sciencedirect.com/science/article/pii/S0167268111001521?casa_token=2k6phzzkLB8AAAAA:x63FRQqFShm_suZYZ_ByHTxC-SYKI_g8I4LQNFsriZIc222jkWd5YeF-iuIUM4YlcWWXmBj6YQEDVQ)

- p.8, l.266: masculine generics ("he" to describe a participants) should not be used

Results

- Unstandardized coefficients should always be presented with standard errors (e.g., in Figure 1)

Discussion

- p.10, l.326-328: While I think I understand the point the authors want to make here, this sentence is rather unclear. From what the authors tested, it cannot be inferred that loss frames used to present ILBRS increase people's sense of efficacy

- p.10, l.340-352: There seems to be some confusion between moderations and mediations
